


# Review article: The growth in compound weather events research in the decade since SREX

Lou Brett[1], Christopher J. White[1], Daniela I.V. Domeisen[2,3], Bart van den Hurk[4,5], Philip Ward[4,5] and Jakob Zscheischler[6,7]

[1] Department of Civil and Environmental Engineering, University of Strathclyde, Glasgow, G1 1XJ, UK
[2] University of Lausanne, Lausanne, Switzerland
[3] Institute for Atmospheric and Climate Science, ETH Zurich, Switzerland
[4] Deltares, Delft, 2629 HV Delft, Netherlands
[5] Institute for Environmental Studies (IVM), Vrije Universiteit Amsterdam, Amsterdam, Netherlands
[6] Department of Compound Environmental Risks, Helmholtz Centre for Environmental Research – UFZ, Leipzig, Germany
[7] Department of Hydro Sciences, TUD Dresden University of Technology, Dresden, Germany

*Correspondence to*: Lou Brett (louise.brett@strath.ac.uk)

**Abstract.** Compound events occur when multiple drivers or hazards combine to create societal or environmental risks. Many

high-impact weather and climate events, such as simultaneous heatwaves and droughts, are compound in nature, leading to more severe consequences than individual events. This review examines the growth of compound event research in the decade since the IPCC Special Report on Managing the Risks of Extreme Events (SREX) in 2012, which built on existing approaches to highlight the need to better understand compound events. A systematic review catalogues 366 peer-reviewed papers published between 2012-22, revealing an annual average increase of 60% of papers across the decade, particularly on

multivariate (co-occurring) events. Most studies focus on Europe, Asia, and North America, with significant gaps in Africa, South America, and Oceania. The review highlights certain modulators, such as the El Niño Southern Oscillation, and selected event types including compound floods and high-temperature low-precipitation events as the most studied in the literature. The review recommends expanding research in underrepresented regions and studying a broader range of typologies, events, and modulators. It also calls for greater cross-disciplinarity and sectoral collaboration to improve the understanding of

compound event impacts and manage the growing risks in a changing climate.

## 1 Introduction

Many high-impact weather and climate events arise from complex interactions between combinations of multiple weather / climate drivers and / or hazards – defined as compound events – which can lead to more (or less) severe impacts than their univariate counterparts (Zscheischler et al., 2018). For example, when extreme heat and low precipitation co-occur, crop yields

can decline due to drought (Zscheischler et al., 2018), potentially affecting global grain prices and food security (e.g., Feng &



Hao, 2020; He et al., 2022). The Intergovernmental Panel on Climate Change (IPCC) Special Report on Managing the Risks of Extreme Events and Disaster to Advance Climate Change Adaptation (SREX) (IPCC, 2012) marked the scientific community's initial effort to begin to coordinate thinking on compound events. It defined compound events as *"(1) two or more extreme events occurring simultaneously or successively, (2) combinations of extreme events with underlying conditions*

*that amplify their impact, or (3) combinations of events that, on their own, are not extreme but lead to an extreme event or impact when combined"* (IPCC, 2012, p. 118). Recognising the potentially high-impact nature of compound events, SREX highlighted the need to better understand compound events and stressed the urgency of understanding the changing frequency and severity of extreme weather in a warming climate.

SREX was a cross-cutting report which built on existing approaches from various disciplines including statistics, hydrology

and engineering. These included Hewitt and Burton (1971) that defined compound hazards as *"...a type of atmospheric hazard where several elements act together above their respective damage threshold, for instance lightning, hail, and wind damage in severe storms".* Other concepts that were brought together included extreme value analysis (Coles, 1999), joint probability and statistical dependence (e.g., Samuels & Burt, 2002; Svensson & Jones, 2005; White, 2007) that had been used in hydrology and coastal science to explore the relationship between (and the compounding impact of) hydrological variables such as river

discharge and sea surge. Additionally, these concepts included copulas – a statistical method for describing the joint probabilities of a multivariate distribution – which have continued to be used and developed (e.g., Bevacqua et al., 2017; Tavakol et al., 2020).

Following the publication of SREX in 2012, the definition of compound events within the climate and meteorology communities has evolved and the discipline has grown significantly. Leonard et al. (2014) identified the SREX definition of

compound events as having imposed artificial boundaries that were neither defined in terms of the physical system, nor did they lead to discrete sets of methodologies for analysis. Instead, Leonard et al. (2014) defined compound events more holistically as *"...an extreme impact that depends on multiple statistically dependent variables or events".* This definition, however, excluded compound events that may have occurred from statistically independent events. Consequently, statistically independent compound events such as successive high precipitation events driven by different atmospheric processes

(Robbins, 2016; Shi et al., 2020; Vanelli et al., 2020) would be excluded using the Leonard et al. (2014) definition. To address this, Zscheischler et al. (2018) refined the definition of compound events to be "…*the combination of multiple drivers and/or hazards that contribute to societal or environmental risk".* This has been widely accepted within the compound events community, cited more than 1,500 times at the time of the present study, and has also been adopted be recent reports of the IPCC (e.g., IPCC 2021).

To aid structuring our thinking about the many possible types of compound events, Zscheischler et al. (2020) presented a typology for compound events comprising four compound event categories: multivariate, pre-conditioned, spatially



compounding and temporally compounding (See Table 1 for definitions). Compound event research also shares similarities with the multi-hazards discipline that explores *"...the selection of multiple major hazards that a given country faces, and the specific contexts where hazardous events may occur either simultaneously, cascading, or cumulatively over time"* (UNDRR, 2017). However, while the multi-hazards discipline explores hydrological, meteorological, climatological and geophysical hazards primarily from a risk perspective, compound event research is generally more grounded in quantifying the interconnections between (climate) hazards and their drivers within the climate sciences (Tilloy et al., 2019; Simmonds et al., 2022).

**Table 1. Definitions of compound event typologies from Zscheischler et al. (2020).**

| Compound event typology | Definition |
|---|---|
| **Multivariate** | Where multiple drivers and/or hazards lead to an impact at the same time in a given location e.g., high winds, low precipitation and high solar radiation contributing to wildfires |
| **Temporally compounding** | Where successive hazards lead to an impact in a given location e.g., flooding followed by further flooding |
| **Spatially compounding** | Where hazards across multiple connected locations cause aggregated impacts e.g., wildfires across multiple locations at a given time |
| **Pre-conditioned** | Where weather and/or climate-driven preconditions aggravate impacts of a hazard e.g., snowfall followed by rainfall |

While it is understood that there have been notable developments in the compound events discipline, to date, no systematic review has been undertaken to synthesise and assess the progress made since SREX was published in 2012. In this study, we address this gap by systematically cataloguing and analysing the number, range and types of compound events studied in the 10 years following the publication of SREX to demonstrate progress and scientific advancements, and to identify gaps and opportunities for the discipline to address in the next 10 years within the context of climate change.

This review is structured as follows. Section 2 details the methods used in this study. Section 3.1 explores the overall advancement of compound event research since SREX in 2012, evidenced in the reviewed literature. Section 3.2 outlines the specific types of events and combinations of modulators, drivers and hazards studied in the reviewed publications, and Section 3.3 explores the sector-specific impacts of compound events identified in the review papers. Section 4 discusses and interprets the findings of the review. Section 5 offers a range of recommendations for the compound event community, and Section 6 presents the final conclusions from the review.





## 2 Methods

### 2.1 Quantitative literature review

A quantitative literature review was undertaken using the Web of Science (Clarivate) database to determine the growth of the
compound events discipline by searching the titles, abstracts and keywords of English language peer-reviewed papers
published between 01 July 2012 and 30 June 2022 (referred to as 2012-22 hereafter). This period was selected to capture the
development within compound event research in the 10 years since the publication of SREX in June 2012 (IPCC, 2012).
Primary search terms such as "compound events / hazards", "cascading events / hazards", "coinciding events / hazards" and
"concurrent events / hazards" were selected based on terminology associated with compound events following Tilloy et al.
(2019) (see Table 2 for full list). The Boolean search term "NEAR/3" was used to allow for a gap of up to three words between
the primary search term and either "hazard" or "event", e.g., "compound X and Y events" would be identified without
"compound" and "event" being directly next to each other. Secondary search terms were then used to reduce the number of
papers from unrelated research fields, these included: AND "risk*" OR "climate" OR "weather" OR "interacting*" OR
"dependence" OR "combination" OR "multivariate" OR "coincidence" OR "trigger" OR "domino" OR "cascade" OR
"interrelation" OR "amplify*" OR "chain". Note that secondary search terms were removed if they were included in the
primary search term.

**Table 2. Primary search terms used for the quantitative literature review. The table shows the number of 'hits' per search term (No. returns), the number of papers filtered as out-of-scope (No. excluded) and the number of relevant papers from each search (No. relevant results). The secondary search term used was always ("risk*" OR "climate" OR "weather" OR "interacting*" OR "dependence" OR "combination" OR "multivariate" OR "coincidence" OR "trigger" OR "domino" OR "cascade" OR "interrelation" OR "amplify*" OR "chain"). NB: secondary search term removed if they were included in the primary search term.**

| Search term | No. returns | No. relevant results |
|---|---|---|
| "compound*" NEAR/3 "event*" OR "compound*" NEAR/3 "hazard*" AND secondary search term | 1,698 | 292 |
| "interacting* NEAR/3 event*" OR "interacting* NEAR/3 hazard*" AND secondary search term | 322 | 3 |
| "cascading* NEAR/3 event*" OR "cascading* NEAR/3 hazard*" AND secondary search term | 336 | 17 |
| "multivariate* NEAR/3 event*" OR "multivariate* NEAR/3 hazard*" AND secondary search term | 10,717 | 8 |





| | | |
|---|---|---|
| "interrelating* NEAR/3 event*" OR "interrelating* NEAR/3 hazard*" AND secondary search term | 3 | 0 |
| "coinciding* NEAR/3 event*" OR "coinciding* NEAR/3 hazard*" AND secondary search term | 78 | 0 |
| "dependent*" NEAR/3 "event*" OR "dependent*" NEAR/3 "hazard* AND secondary search term | 2,168 | 0 |
| "triggering* NEAR/3 event*" OR "triggering* NEAR/3 hazard*" AND secondary search term | 1,435 | 4 |
| "amplifying*" NEAR/3 "event*" OR "amplifying*" NEAR/3 "hazard*" AND secondary search term | 34 | 0 |
| "combined*" NEAR/3 "hazard*" OR "combined*" NEAR/3 "event*" AND secondary search term | 2,574 | 12 |
| "concurrent*" NEAR/3 "hazard*" OR "concurrent*" NEAR/3 "event*" AND secondary search term | 478 | 30 |
| "multihazard*" OR "multi-hazard*" AND secondary search term | 997 | 23 |
| DAMOCLES publications list- Damocles.compoundevents.org | 79 | 79 |
| **Total studies:** | **20,919** | **366** (468 total with 102 repeats) |

Despite the use of a range of primary and secondary search terms in an attempt to develop a comprehensive set of key search terms for the paper appraisal, it cannot be guaranteed that all relevant search terms to capture compound event research were

included due to the extensive and varied terminology used within the scientific community. Papers collated by the DAMOCLES Cost Action Compound Research Publication List (CompoundEvent.org) were, therefore, also incorporated into the paper appraisal to include other relevant research papers not captured by the search terms employed in the study. Papers highlighted by the above searches were filtered for relevance, excluding papers from unrelated fields such as chemistry, oncology, and astronomy. Of the 20,919 returns, 366 papers were within the scope of this review (Table 2).

The 366 identified papers were then grouped by region and by year of study to determine any geographical and temporal patterns. The months to determine each year were set from July to June as this is when SREX was published (IPCC, 2012),





e.g. 2014-2015 represents July 2014 to June 2015. The conventional division of continents was used to group regional analysis, i.e., Africa, Asia, Europe, North America, Oceania, and South America. No papers explicitly analysed Antarctica; thus, it was excluded. The regional groupings were based on the location of compound events being analysed as opposed to institutional

publishing. For example, if a European institution studied compound events in China, the paper was included in the Asia count. Studies spanning several continents, or analysing events over an ocean, were grouped as multi-regional.

Theoretical research, review papers, frameworks and other relevant papers that did not focus on place-specific analysis of compound events are classified as 'other'. These 55 'other' papers are included in the review to show the overall development of compound event research throughout the 10-year review period.

**2.2 Quantifying the advancements of compound events science**

Out of the 366 papers, 55 did not quantitatively analyse any compound events. In addition, several papers examined more than one compound event, such as Ridder et al. (2020) which studied 27 hazard papers. Consequently, this review catalogued 388 separate compound events from across the 366 papers reviewed.

By examining the methodologies of the reviewed papers, each separate compound event was categorised into four typologies

following Zscheischler et al. (2020) (see Table 1 for definitions). This typology was developed to aid compound event analysis by facilitating the selection of appropriate modelling tools and analysis. However, it is acknowledged that the boundaries between the four groups are not precise (Zscheischler et al., 2020). Consequently, several compound events were categorised as a combination of typologies.

Following the typology categorisation, modulators, drivers, and hazards were catalogued by examining each paper's text and

applying expert judgement accordingly (see Table 3 for definitions). Modulators – modes of climate variability such as such as El Niño Southern Oscillation (ENSO), the North Atlantic Oscillation (NAO) or a persistent atmospheric blocking – can influence both the frequency and location of climate drivers like tropical cyclones or storm fronts, affecting the frequency and/or intensity of hazards (Zscheischler et al., 2020). Some papers were found to directly refer to the type of climate variability (e.g., ENSO) without using terms like "modulator" or "mode of variability." Consequently, after an initial search for these

terms, expert judgement was used to identify additional modulators. These were then grouped into two categories: (1) modulators quantitatively analysed and (2) modulators mentioned but not analysed. Modulators were further classified into (1) ocean warming patterns and (2) persistent atmospheric Rossby wave configurations to compare their frequency of mention or analysis in the reviewed papers.






**Table 3. Definitions of compound event components from Zscheischler et al. (2020) used in this study.**

| Compound event component | Definition |
|---|---|
| **Hazard** | Climate-related phenomena that have the potential to cause an impact, and can include events such as floods, strong winds, frost, precipitation, heatwaves, droughts, and wildfire; the hazard does not need to be extreme itself, provided it triggers (or could trigger) an impact |
| **Driver** | Hazards are caused by one or more climate driver(s), which can include weather systems, such as tropical cyclones, severe storms, or stationary high-pressure systems |
| **Modulator** | Large scale climate system states considered to be the 'drivers of drivers', for instance low-frequency modes of climate variability such as El Niño–Southern Oscillation that can influence both the frequency and location of drivers, affecting in turn the frequency and/or intensity of hazards |
| **Impact** | The impact of the hazards, for example flood damage, crop damage, human health, or energy outages |

Compound events involve the combination of drivers and/or hazards (Zscheischler et al., 2018). While these can be grouped separately, they often share overlapping indicators or variables. For instance, in a pre-conditioned event where spring
conditions influence summer vegetation growth, drivers (high spring temperatures, low precipitation, high radiation, evapotranspiration, and increased plant growth) share commonalities with the hazards (high summertime temperature and low precipitation) (Bevacqua et al., 2021a). Consequently, to highlight the complex combinations of drivers and hazards studied over the 10 years since SREX, expert judgement was employed in some cases to help catalogue the variables and indicators representing both drivers and hazards together, referring to these as hydrometeorological variables.

**2.3 Impacts of compound events**

The sector-specific impacts of each study, mentioned within the text of reviewed papers, were identified and recorded. These impacts were then grouped into 10 sector-specific categories and recorded to illustrate the range of impacts that the reviewed papers have considered over the 10 years since SREX. If a paper mentioned multiple impacts (e.g., agricultural production and ecosystem health), each impact was recorded separately.





## 3 Results

### 3.1 Growth of compound event research

The discipline of compound events has seen significant growth in the decade since the release of SREX in 2012, evidenced through a rapidly increasing number of publications (Fig.1). Except for 2014-15, the number of published papers included within this review rose year-on-year, from fewer than 20 papers annually before 2018 to 116 papers in 2021-22. From 2013 to 2022, there was an average annual increase of 60% in the number of compound events papers. This contrasts with the 4% annual average increase in the number of published peer-reviewed science and engineering journal articles and conference papers between 2009 and 2019, reported by the National Science Board (2019). Several factors may have influenced the growth of peer-reviewed compound events papers, such as the raised profile via SREX (IPCC, 2012), the consolidation of the definition of compound events (Zscheischler et al., 2018), and research initiatives such as the European COST Action DAMOCLES.

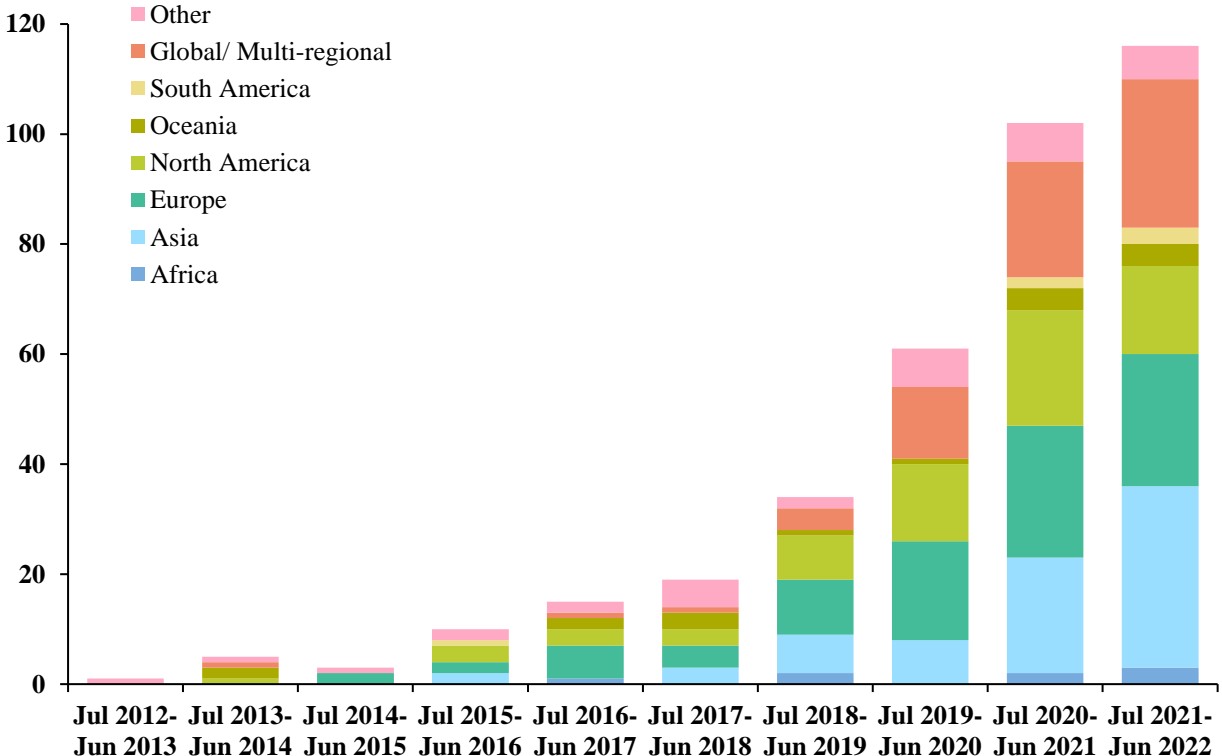

**Figure 1. Number of papers published per year (Jul-Jun) and the regional focus of the research from 2012 to 2022. Where papers focus on two or more continents, the result is classified as 'multi-regional'. Where papers are not place-based research, they are classified as 'Other'.**





Geographically, results show that Europe was the most studied region, with 90 papers focusing on European locations (Fig. 1). Asia was the second most studied region and North America the third, with 74 and 69 papers respectively. There was a relatively consistent growth in the annual number of papers within Europe and North America. In contrast, the annual number of papers focusing on Asia rapidly grew between 2019-2021 with a doubling of publications in this short window. Similarly, a rapid increase in the number of multi-regional studies was found, increasing from four papers in 2018-19 to 27 papers in

2021-22. Conversely, the number of publications remained relatively low for studying Oceania, Africa and South America with 17, eight and six publications respectively throughout the last ten years. This disparity could be related to factors such as limited research funding, data availability or access, or awareness of compound events (Jacobs et al., 2016; Overland et al., 2022).

### 3.2 Advancements of compound event science

#### 3.2.1 Categorisation of compound events

Fig. 2 categorises the compound events studied within the reviewed papers following the Zscheischler et al. (2020) typology (refer to Table 1 for definitions). The results show that multivariate events were the most studied category, encompassing 71.9% of the compound events. Multivariate events were analysed over six times more than the second most analysed category, temporally compounding events, which accounted for 11.9% of the studies. Research analysing spatially compounding and

pre-conditioned events accounted for a small proportion of studies, at 3.6% and 2.8%, respectively. Events that were categorised as a combination of compound event typologies accounted for the remaining 9% of events.

For the 9% of events that were categorised as a combination of typologies, 3.9% were classified as both pre-conditioned and temporally compounding, including wildfires followed by high precipitation (HighP) events (e.g., Jacobs et al., 2016). Additionally, 2.6% of events were classified as multivariate and spatially compounding, including high temperature (HighT)

and low precipitation (LowP) events co-occurring across multiple regions (e.g., Feng & Hao, 2020). 1.5% of events were classified as multivariate and temporally compounding, including Van der Velde et al. (2020), who explored how a warm autumn, then warm and wet spring, led to wheat production loss in France in 2016; 1.0% were classified as pre-conditioned and multivariate, including Flach et al. (2018), who explored how vegetation modulates the impact of climate extremes on gross primary production; and 0.8% of events were classified as temporally compounding, pre-conditioned, and multivariate,

including Bastos et al. (2020), who studied the direct and seasonal legacy effects of the 2018 heatwave and drought on European ecosystem productivity.

Overall, these results show that a wide variety of compound event typologies have been explored; however, multivariate studies have dominated compound event research in the decade since SREX.



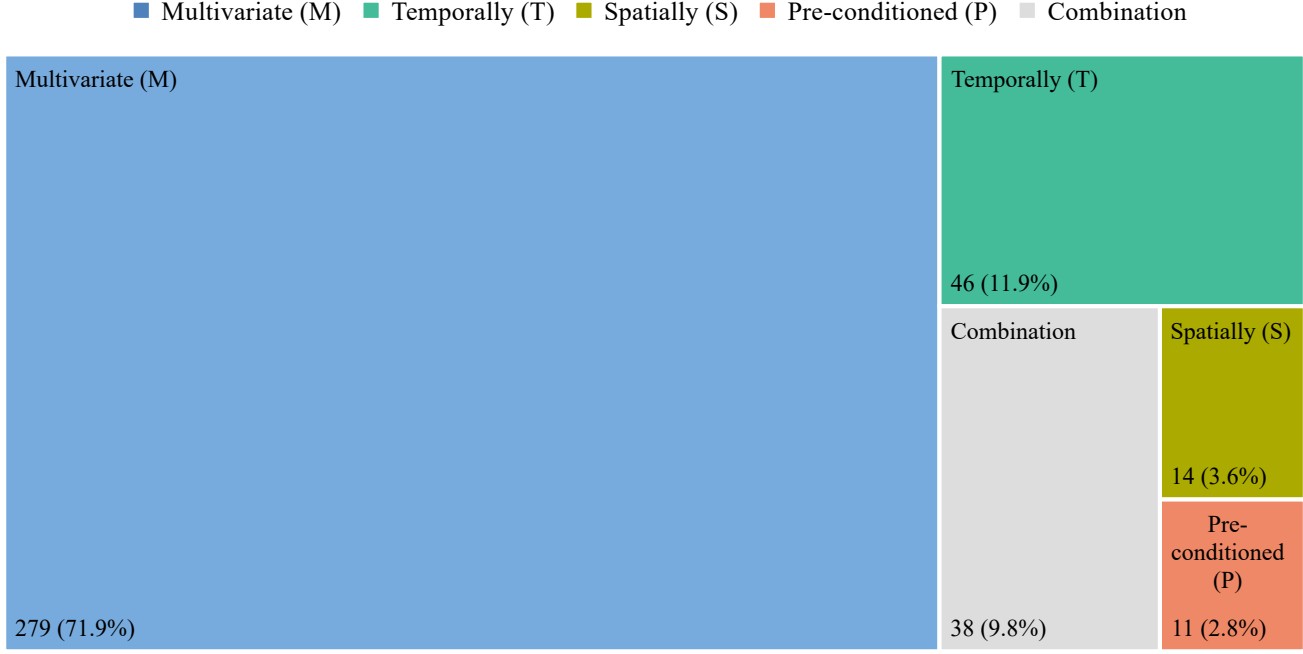


**Figure 2. Count of the number of studies reviewed for each compound event category, and their combinations (in grey) (total = 388 events). The percentage (%) indicated is the proportion of all compound event types studied.**

### 3.2.2 Compound event modulators

Modulators can influence both the frequency and location of compound event drivers, thus affecting the frequency and/or
intensity of hazards (see Table 3 for full definition). Twenty-four different modulators were either mentioned or analysed
across 14.8% of the reviewed papers (see Table 4 for the full list). Of these 24 modulators, 17 types were mentioned within
reviewed papers for their influence on the frequency and/or intensity of compound events across 12.2% of reviewed papers.
For example, Robbins et al. (2016) explore landslide-triggering precipitation events using satellite precipitation estimates,
highlighting how larger-scale variations such as the El Niño Southern Oscillation (ENSO) and Madden-Julian Oscillation
(MJO) can influence seasonal rainfall without specifically analysing the influence of these modulators on precipitation events.
Furthermore, 15 types of modulator were analysed for their influence on the frequency and/or severity of compound events
across 8.2% of the reviewed papers. For example, Hao et al. (2018) quantify the relationship between compound dry and hot
events and ENSO at a global scale, and Jarvis et al. (2018) explore the influence of ENSO and the Indian Ocean Dipole (IOD)
on winegrape maturity in Australia. Consequently, modulators were mentioned across 4% more of the reviewed papers than
they are analysed for their influence on compound event drivers and/or hazards.





The same three types of modulators were both the most mentioned and the most analysed (see Table 4). ENSO was the most studied modulator, mentioned in 4.9% of the reviewed papers and analysed in 6%. ENSO was analysed over three times more than the second most studied modulator, the North Atlantic Oscillation (NAO), which was mentioned in 1.9% of papers and analysed in 1.6% of papers. For example, Hiller & Dixon (2020) study how compound winter storms, compound wind and rain events, are more likely to occur in northern Europe when the NAO is in a positive phase. The other most studied modulators included the Pacific Decadal Oscillation (PDO), mentioned in 0.8% of reviewed papers and analysed in 1.6%; the Atlantic Multi-decadal Overturning circulation (AMO), mentioned in 0.3% of reviewed papers and analysed in 1.1%; and the Indian Ocean Dipole (IOD), mentioned in 0.5% of papers and analysed in 0.8% of papers.

When comparing the two categories of modulators, 'Modulators Driven by Ocean Warming Patterns' (ocean-driven modulators) were both mentioned more frequently and analysed more extensively in the reviewed papers than 'Modulators Related to Persistent Atmospheric Rossby Wave Configurations' (atmospheric-driven modulators). Specifically, ocean-driven modulators were mentioned 25 times and analysed 38 times, while atmospheric-driven modulators were mentioned 18 times and analysed 14 times. Although, overall, 24 more papers analysed ocean-driven modulators compared to atmospheric-driven modulators, fewer types of ocean-driven modulators were analysed (7 compared to 8 types of atmospheric modulators). Notably, only one type of oceanic modulator was mentioned without being analysed, in contrast to 8 types of atmospheric modulators that were mentioned but not analysed.

Nine modulators were mentioned within the reviewed studies but not analysed. These include the Interdecadal Pacific Oscillation (IPO), which was mentioned as a possible modulating influence for decadal variability in rainfall and temperature over southeast Australia (Kirono et al., 2017), and the North Pacific Oscillation (NPO), where boreal wintertime variation in NPO was mentioned as having a significant correlation with austral spring rainfall anomalies (Steptoe et al., 2018).

**Table 4. List of climate modulators outlined within reviewed papers, and the count that each modulator was either qualitatively mentioned or quantitatively analysed. Modulators are grouped into two categories: (1) modes driven by ocean warming patterns, and (2) modes related to persistent atmospheric Rossby wave configurations.**

| Modulator | No. mentions | Papers mentioned | No. analysed | Papers analysed |
|---|---|---|---|---|
| **Modes Driven by Ocean Warming Patterns:** | | | | |
| El Niño-Southern Oscillation (ENSO) | 18 | Liu et al., 2015; Robbins, 2016; Tencer et al., 2016; Kirono et al., 2017; Steptoe et al., 2018; Zscheischler & Seneviratne, 2017; Herdman et al., 2018; Mukherjee et al., 2018; Zhou et al., 2019; Hao & Singh, 2020; Hillier & | 22 | Hao et al., 2018; Jarvis et al., 2018; Liu et al., 2018; Zhou & Liu, 2018; Anderson et al., 2019; Hao et al., 2019; Hao et al., 2020a; Wu & Leonard, 2019; De Luca et al., 2020; Hao et al., 2020b; Mukherjee et al., 2020; Shi et al., 2020; Dykstra & |





| | | Dixon., 2020; Ridder et al., 2020; Yu & Zhai, 2020; Zscheischler et al, 2020; Ballarin et al., 2021; Wu et al., 2021b; Li et al., 2022; Zamora-Reyes et al., 2022. | | Dzwonkowsi, 2021; Feng et al., 2021; Hao et al., 2021b; Le Grix et al., 2021; Singh et al., 2021; Wu et al., 2021b; Zhang et al., 2022; Camus et al., 2022; Fish et al., 2022; Richardson et al., 2022. |
|---|---|---|---|---|
| Pacific Decadal Oscillation (PDO) | 3 | Yu & Zhai, 2020; Abatzoglou et al., 2021; Wu et al., 2021a. | 6 | Hao et al., 2020a; De Luca et al., 2020; Mukherjee et al., 2020; Le Grix et al., 2021; Li et al., 2021b; Wu et al., 2021b. |
| Atlantic Multi-Decadal Overturning (AMO) | 1 | Yu & Zhai, 2020. | 4 | De Luca et al., 2020; Dykstra & Dzwonkowski, 2021; Li et al., 2021c; Wu et al., 2021. |
| Indian Ocean Dipole (IOD) | 2 | Steptoe et al., 2018; Ridder et al., 2020. | 3 | Le Grix et al., 2021; Singh et al., 2021; Zhang et al., 2020. |
| Tropical North Atlantic (TNA) | | | 1 | Singh et al., 2021. |
| North Pacific Gyre Oscillation (NPGO) | | | 1 | Le Grix et al., 2021. |
| El Nino Modoki | | | 1 | Le Grix et al., 2021. |
| Interdecadal Pacific Oscillation (IPO) | 1 | Kirono et al., 2017. | | |
| **Sub-total:** | **25** | | **38** | |
| **Modes Related to Persistent Atmospheric Rossby Wave Configurations:** | | | | |
| North Atlantic Oscillation (NAO) | 7 | Sedlmeier et al., 2018; Steptoe et al., 2018; Ridder et al., 2018; Hao & Singh, 2020; Messmer & Simmonds, 2021; van der Wiel et al., 2021; Wu et al., 2021b. | 6 | Hillier et al., 2020; Hillier & Dixon, 2020; Mukherjee et al., 2020; Le Grix et al., 2021; Wu et al., 2021b; Camus et al., 2022. |
| Madden- Julian Oscillation (MJO) | 1 | Robbins, 2016. | 2 | Anderson et al., 2019; Cowan et al., 2022. |
| Pacific North American Pattern (PNA) | 1 | Steptoe et al., 2018. | 1 | Camus et al., 2022. |
| Arctic Oscillation (AO) | 1 | Steptoe et al., 2018. | 1 | Shi et al., 2020. |
| East Asia Pacific Pattern | | | 1 | Wang & Wang, 2018. |
| Western European Pressure Anomaly (WEPA) | | | 1 | Camus et al., 2022. |





| Antarctic Oscillation Index (AOI) | | | 1 | Le Grix et al., 2021. |
|---|---|---|---|---|
| Silk Road teleconnections (SR) | | | 1 | Wang & Wang, 2018. |
| North Pacific Oscillation (NPO) | 1 | Steptoe et al., 2018. | | |
| East Atlantic Pattern | 1 | Ridder et al., 2018. | | |
| Southern Annular Mode (SAM) | 1 | Steptoe et al., 2018. | | |
| Scandinavian Pattern (SCP) | 1 | Steptoe et al., 2018. | | |
| South Atlantic Convergence Zone (SACZ) | 1 | Ballarin et al., 2021. | | |
| Blocking events | 1 | Horton et al., 2016. | | |
| High pressure systems | 1 | Bevacqua et al., 2021a. | | |
| Storms | 1 | Bevacqua et al., 2021a. | | |
| **Sub-total** | **18** | | **14** | |
| **Total:** | **43** | | **52** | |

### 3.2.3 Hydrometeorological drivers and hazards

Table 5 lists the combinations of hydrometeorological variables that were studied in at least two of the reviewed papers, demonstrating the range of compound events studied in the 10 years following SREX. These combinations were also categorised into the compound event typologies to further explore the range of events studied. A full list of compound event combinations can be found in the supplementary material.

**Table 5. All compound drivers and/or hazard components analysed more than once within the reviewed papers. The letters in the column headers represent multivariate (M), pre-conditioned (P), temporally compounding (T), spatially compounding (S) and combinations (combo) of compound event types. The numbers represent the number of times each driver and/or hazard combination occurred. Definitions: High temperatures during day (HighT), low temperatures (LowT), low precipitation (LowP), high precipitation (HighP), sea surge (Surge), high wind (Wind), river discharge (RiverD), high sea level (HSL), high temperatures during night (HighTnight ), low soil moisture (LowSM), high soil moisture (HighSM), wave height (Wave), snow (Snow), high relative humidity (HighRH), wildfire (Wildfire), high pressure (HighPres), flood (Flood), groundwater levels (Groundwater), vapour pressure deficit (VPD), tropical cyclone (TC), landslide (Landslide).**

| Components | M | P | T | S | Combination |
|---|---|---|---|---|---|
| 1. HighT and/then LowP | 76 | | 3[a] | | 9[b] |
| 2. HighT and/then HighP | 18 | | 7[c] | | |
| 3. HighP and Surge | 22 | | | | |





| | | | | | |
|---|---|---|---|---|---|
| 4. HighP and Wind | 18 | | | | 1 |
| 5. RiverD and Surge | 17 | | | | |
| 6. LowP and/then HighP | | | 11[d] | 2 | |
| 7. HSL & RiverD | 10 | | | | |
| 8. LowT and HighP | 10 | | | | |
| 9. HighP and/then HighP | | | 7 | 2 | |
| 10. HighT and HighT$_{night}$ | 9 | | | | |
| 11. LowSM and/then HighT and LowP | 7 | | | | 2 |
| 12. LowP and LowT | 8 | | | | |
| 13. LowSM and HighT | 8 | | | | |
| 14. Surge and Wave | 5 | | | | |
| 15. Surge and Wind | 5 | | | | |
| 16. LowP and/then LowP | | | 1 | 4 | |
| 17. HSL and HighP | 4 | | | | |
| 18. Snow then HighP | | 4 | | | |
| 19. HighSM then HighP | | 4 | | | |
| 20. HighT and HighRH | 4 | | | | |
| 21. Wildfire then HighP | 4 | | | | |
| 22. HighT and HighPres | 4 | | | | |
| 23. RiverD and RiverD | | | | 3 | |
| 24. LowT and HighP and Wind | 3 | | | | |
| 25. HighP and RiverD | 2 | | | | |
| 26. Flood and Wind | 2 | | | | |
| 27. HighP and HSL and Groundwater | 2 | | | | |
| 28. HighT and LowP and VPD | 2 | | | | |
| 29. TC then HighT | | | 2 | | |
| 30. HighP then Landslide | | | 2 | | |
| 31. LowP then LowSM | | | | | 2 |

Notes:
a. One out of the three temporally compound events LowP then HighT.
b. Eight out of the nine combinations of HighT and LowP spatially/multivariate for crop yields, one out of nine is HighT and LowP repeatedly and legacy on plants (M/T/P).
c. Two out of the seven temporally compounding events HighP then HighT.
d. Two out of the 11 temporally compounding events are HighP then LowP.

Results show that high temperature (HighT) and (or followed by) low precipitation (LowP) events were the most studied single combination of hydrometeorological variables. This combination was analysed 88 times, accounting for 22.7% of all





compound events analysed within the reviewed papers. Multivariate events accounted for 76 of the 88 HighT and LowP events analysed, including Singh et al. (2022) who evaluate joint projections of temperature and precipitation extremes across Canada. Three HighT and LowP events were temporally compounding events, including Weber et al. (2020) analysing HighT extremes followed by LowP extremes in Africa, and nine HighT and LowP belonged to the class of combined typologies. HighT and LowP events were studied over three times more than the second most studied combination, HighT and/then high precipitation

(HighP) analysed 25 times.

Of the 25 HighT and/then HighP events, 18 were multivariate and seven were temporally compounding. The sequence of the seven temporally compounding HighT and HighP events varied. Five HighT then HighP events were analysed, including Das et al. (2022), who explored population exposure to compound extreme events in India. Two HighP then HighT events were analysed, including Chen et al. (2021), who detected increases in sequential flood and heatwave events across China.

Alongside compound hot events like LowP and HighT or HighT followed by HighT at night, compound events related to flood conditions were frequently studied. These include events with at least one variable such as HighP, Surge, or River discharge (RiverD). For instance, HighP and Surge events were the third most studied combination, explored 22 times and accounting for 5.7% of events, all categorised as multivariate. Zhang et al. (2020), for example, integrates traditional hydrologic and hydrodynamic models into a single platform to simulate compound floods from coastal storm surge and precipitation-induced

river flooding together. Other combinations of hydrometeorological variables studied more than 10 times include HighP and Wind (19 events), RiverD and Surge (17 events), LowP and/then HighP (13 events), High Sea Level (HSL) and RiverD (10 events), and Low Temperature (LowT) and HighP (10 events). These combinations involve multiple variables related to compound flooding.

Exploring the individual hydrometeorological variables in Table 5 (full list of variables studied available in supplementary

information), three variables featured in over 39% of compound event studies. HighT was studied in 49.3% of events, making it the most studied single variable. For example, Collins (2021) explores how the frequency of compound hot-dry extremes in Australia has changed since 1889. LowP was the second most studied variable, and HighP was the third, studied in 40.1% and 39.9% of events respectively. He & Sheffield (2020) explore both LowP and HighP variables, studying lagged drought-pluvial seesaw occurrence globally. Although HighT was the single most studied hydrometeorological variable, at least one variable

related to compound flooding (e.g., HighP, RiverD, Surge, HSL) was studied in 54.6% of the events displayed in Table 5, which accounts for 43% of all events studied in the reviewed papers. This highlights that compound flood events were researched more than compound hot and dry events, using a range of combinations of hydrometeorological variables to explore compound flooding.



While results can be used to highlight the combinations of hydrometeorological variables that have been studied within the
reviewed papers, we can also begin considering what is missing, or has been less studied, throughout the 10 years since SREX.
For example, while LowP and HighP were studied in a similar number of events, HighT was studied over seven times more
than LowT, which was studied in 6.9% of events displayed in Table 5. Furthermore, a range of other hydrometeorological
variables such as high soil moisture (HighSM), relative humidity, groundwater level, cloud coverage, and solar irradiance were
less studied throughout the reviewed papers. Supplementary information provides a full list of compound events studied within
the reviewed papers, including all the combinations of hydrometeorological variables that were only studied once.

Overall, these results highlight the combinations of hydrometeorological variables that have been studied more than once
within the reviewed papers throughout the 10 years since SREX. These results show that HighT & LowP events were the
single most studied combination of hydrometeorological variables, whilst combinations of variables related to compounding
flooding were collectively analysed more than HighT and LowP events. These results also indicate a wide range of
hydrometeorological variables that have been studied relatively less in the 10 years since SREX.

### 3.3 The impacts of compound events

Many of the reviewed papers include references to the wider sector-specific impacts of their research. In some cases, impacts
such as agricultural yields, mortality or wider socio-economic or environmental indicators have been quantitatively analysed
(e.g., Feng et al., 2021; Bastos et al., 2021); in others, researchers qualitatively highlighted the relevance of their research to
wider impacts (e.g. De Luca et al., 2017; Lesk & Anderson, 2021). This review explores the quantitatively analysed and
qualitatively highlighted sector-specific impacts together to gain an overall understanding of the types of impacts that
researchers are considering within compound event research (Fig.3).

The results indicate that flood/storm damage was the most frequently referenced impact in the reviewed papers, highlighted in
96 studies. This supports the findings in section 3.2.3, which emphasise that compound flooding has been relatively well-
studied over the past 10 years since SREX. For example, De Luca et al. (2017) qualitatively describe the damages and broader
disruptions caused by flooding before analysing the relationship between extratropical cyclones and multi-basin, spatially
compounding flooding. Similarly, Tanir et al. (2021) quantitatively analyse the socio-economic vulnerability resulting from
compound flooding in Washington, DC, by combining flood exposure with a socio-economic vulnerability index to identify
at-risk populations.

Agricultural production was the second most referenced impact, highlighted in 79 studies. For example, Lesk & Anderson
(2021) highlight how extreme heat and drought often reduce yields of important food crops, putting stress on regional and
global food security. Several of the multivariate and spatially compounding HighT and LowP events also quantitatively analyse
breadbasket failures from co-occurring drought on crop yields (e.g., Potopova et al. 2020 & Feng et al. 2021). Ecosystem





health was highlighted in 69 studies. For example, Vogel et al. (2021) highlights the socio-economic and ecological impacts
of compound warm spells and drought conditions on ecosystem health and wider biodiversity in the Mediterranean.
Furthermore, Bastos et al. (2021) quantify changes in vegetation vulnerability from two compound dry and hot summers in
2018 and 2019 by studying the enhance vegetation index (EVI) anomalies. Health and mortality impacts were highlighted in
61 studies. For example, Wang et al. (2021) qualitatively highlights the health-related impact of anthropogenic emissions and
urbanisation on compound heat extremes; and Plavcová & Urban (2020) quantitatively analyse the intensified impacts of
320 compound winter extremes, relative to single hazards, on mortality rates in Czech Republic. A wider range of impacts such as
infrastructure damage, energy infrastructure and markets, water resource management, insurance losses and landslide damage
were each highlighted in < 20 studies.

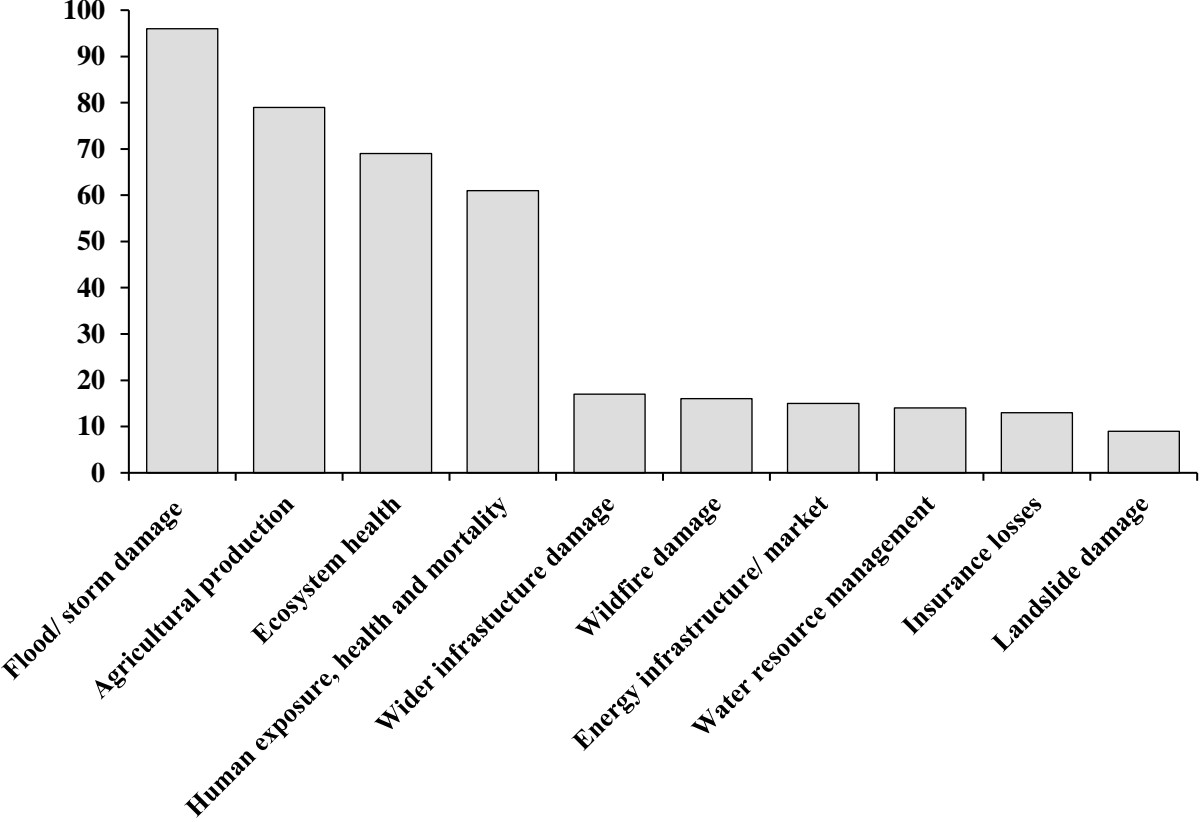

**Figure 3. Count of references to sector-specific impacts of compound events in the reviewed papers. If multiple sector-specific**
**impacts were referenced within a single study, each of the impacts has received a 'count'.**





## 4 Discussion

This review paper catalogued 366 studies on compound events published in the decade since the SREX report, revealing a notable 60% annual increase in such publications. Several factors likely contributed to this growth. One catalyst was likely the high-profile IPCC (2012) report, which highlighted the importance of studying compound events in the context of extreme weather. Additionally, the perspective paper by Zscheischler et al. (2018) provided a clearer definition of compound events, gaining significant support, especially within climate science. Furthermore, the emergence of major initiatives such as the Risk-KAN community and the European COST Action DAMOCLES (CA17109) likely contributed to the increase in compound event publications and played a role in fostering stronger research communities within the discipline.

The results demonstrate significant regional disparities in compound event analysis, with 64% of the studies focused on Europe, Asia, or North America, while only 8.5% of studies concentrated on Africa, South America, or Oceania. The predominance of Europe and North America as frequently studied regions can be attributed to their significant share of global academic research (Kamalski & Plume, 2013). The 'global north' generally benefits from more research funding, opportunities, data availability, and access (Jacobs et al., 2016; Overland et al., 2022), likely contributing to the larger research output within these regions.

Publications focused on Asia have notably increased since 2017, about 3-4 years after similar growth in Europe and North America. This rise in publications may be attributed to specific research groups in Asia beginning to focus on compound events, such as one group that published seven papers on hot dry events in China between 2019 and 2022 (e.g., Wu et al., 2019; Wu & Jiang, 2022). Additionally, global and multi-regional studies have grown significantly since 2018, likely driven by global-scale model ensembles like CMIP5 and CMIP6, and satellite data enabling large-scale studies (e.g., Zhou et al., 2019; Wu et al., 2021; Bevacqua et al., 2021b). However, large geographical gaps remain, particularly in South America, Oceania, Africa, and the oceanic and high-latitude regions, which are highly understudied (e.g., Le Grix et al., 2021). Expanding studies in these areas would improve physical risk estimates. Satellite or reanalysis datasets could address these gaps, as seen in Jacobs et al. (2016), where Google Earth images and field observations detected flash floods in the data-poor Rwenzori Mountains, Uganda.

Additionally, results show a significant disparity in the types of compound events studied, with multivariate events examined over six times more often than other types. Several factors likely contribute to the high prevalence of multivariate studies. Firstly, the study of multivariate events may be more intuitive for researchers compared to other compound event categories. For example, pre-conditioned or temporally compounding events require consideration of additional factors, such as the time lag between hydrometeorological variables and their biological effects on plant development, highlighting the need for further research and deeper understanding (e.g., Khanal et al., 2019; Matusick et al., 2018; Bastos et al., 2020).





Furthermore, established statistical methods for understanding joint probabilities, such as the use of copulas, are frequently used to explore the relationship between hydrometeorological variables within compound event studies (e.g., Bevacqua et al., 2017; Couasnon et al., 2018; Naseri & Hummel, 2022). Several studies, such as those by Ridder et al. (2020) and Sutanto et al. (2020), also analyse multiple multivariate combinations in a single study, further boosting the proportion of multivariate analysis. Overall, the predominance of multivariate event studies can be attributed to their intuitive nature, available statistical methods, and the number of papers that have examined multiple multivariate combinations.

Understanding complex interactions between drivers and hazards across all compound event types is essential for accurate risk assessment and mitigation (Zscheischler et al., 2018; Raymond et al., 2020; van den Hurk et al., 2023). This is particularly important for temporally compounding events, where the timing and sequence of events play a critical role in their overall impact (Leonard et al., 2014). Furthermore, understanding spatially compounding events is important for regional risk assessments, helping identify areas affected simultaneously and improving disaster response (Hillier et al., 2015). Additionally, studying pre-conditioned events can enhance early warning systems by recognising conditions that precede severe impacts, potentially reducing the risk to human life and property (Singh et al., 2021).

Modulators of compound events were mentioned across 12.2% of the reviewed papers and analysed in 8.2%, with ENSO emerging as the most researched modulator. Understanding the effects of modulators on compound event occurrence and severity is important for predicting weather variations like droughts, floods, and temperature extremes (Hao et al., 2020a), aiding in planning and mitigating impacts on food security and ecosystems (Jarvis et al., 2018). Furthermore, enhanced knowledge of how modulators affect weather patterns can improve early warning systems (EWSs), allowing for better preparedness and risk reduction in areas prone to extreme weather, thereby giving communities with more time to implement preventive measures (Hillier et al., 2015; Singh et al., 2021).

Despite research being conducted on a limited number of modulators, there are opportunities to explore the influence of a wider range of modulators on compound events. Furthermore, few studies have explored how combinations of modulators influence extreme weather occurrence (Singh et al., 2021). Thus, future research could analyse both individual and combined modulator effects on compound event occurrence and severity. Expanding this focus could provide a more comprehensive understanding of the influence of modulators on compound event occurrence.

HighT and LowP events were the most studied combination of hydrometeorological variables, accounting for 22.7% of analysis; this is likely due to the growing impact of heat events, exacerbated by climate change (IPCC, 2021). HighT and LowP events, for instance, can severely impact agriculture, ecosystems, and human health by worsening drought conditions (Flach et al., 2018; Lesk et al., 2021; Das et al., 2022). However, multiple combinations of variables related to compound flooding were studied even more extensively than HighT and LowP, representing > 43% of the compound events reviewed. This focus



on flood-related studies may be attributed to the long-standing history of research into hydrological joint probability and statistical dependence (e.g., Svensson & Jones, 2005; White, 2007).

Low-temperature (LowT) compound events, which can result from high-pressure – dominated periods in winter or low-pressure systems outside winter, also have significant socio-economic impacts such as increased heating costs, higher elderly mortality rates, transport disruption, and reduced renewable energy production (Plavcová & Urban, 2020; Hillier & Dixon, 2020; Thornton et al., 2017). While LowT events will still occur in a warming climate, they may have different characteristics, such as more frequent LowT and HighP events in winter (De Luca et al., 2020). Despite their potential impacts, LowT events were studied over seven times less frequently than HighT events (**Table 5**).

Furthermore, emergent compound hazards such as HighT or LowT combined with cloud cover or solar radiation can significantly impact renewable energy demand and production (Thornton et al., 2017; Van der Wiel et al., 2021). The limited research on these hazards, e.g., low wind and cloud coverage, may be due to factors like limited data availability and suitability. For instance, the spatial resolution of general circulation models (GCMs) is often inadequate for site-specific wind climate analyses (Cradden et al., 2012); and reanalysis datasets, particularly ERA5, also tend to show uncertainties in wind speeds over mountainous and coastal areas, where they can be significantly under- and overestimated, respectively (Gualtieri, 2022). Additionally, different terminology used to discuss compound events might have resulted in relevant research being undiscovered for this review. For instance, in the renewable energy sector, terms like 'dunkelflaute' describe low wind and high cloud coverage compound events (Li et al., 2021), without referencing compound events or related synonyms. Promoting uniform terminology across disciplines could foster collaboration and the sharing of best practices and methodologies. This collaboration could enhance our understanding of emerging compound events and improve our capacity to manage their impacts in a rapidly changing world.

Flood and storm damage (96 references) was the most cited impact of compound events, aligning with the findings that hydrometeorological variables related to compound flooding were the most researched. Agricultural production (79 references) and ecosystem health (69 references) were also frequently mentioned. For instance, Apel et al. (2016) used high-resolution RapidEye satellite data to assess building resistance to urban flooding, while English et al. (2017) applied the American Society of Civil Engineers (ASCE) standards to mitigate flood and wind damage from hurricanes. Additionally, a mixture of global datasets, such as maize yields available from the Food and Agriculture Organisation (FAO) (e.g., Feng et al., 2019; Feng & Hao, 2020), and regional datasets, such as provincial yields of wheat and barley in Spain (e.g., Ribeiro et al., 2020), were used to explore the impacts of compound events on agricultural production. Data availability, access, and collaboration with specific organisations could have influenced the extent to which these different impacts were referenced.





By expanding networks to encompass a wider range of sectors, such as disaster risk management, transport, and telecommunications, and fostering greater integration across disciplines, including multi-hazard research, applied meteorology, engineering, and sustainable development, we could greatly enhance the compound event research network. This integration can facilitate the sharing of research and insights across fields, broadening our understanding of the varied impacts of compound events. For instance, studies have already investigated the effects of combined weather events on renewable energy supply and demand, as well as the management of offshore energy production facilities (Thornton et al., 2017; Ren et al., 2021). However, these studies were not included in this review due to the different terminology employed. Standardising terminology and fostering collaboration between scientists, policymakers, and practitioners, can improve the capturing of compound event impacts and encourage research on issues relevant to practitioners. This can lead to more comprehensive risk assessments, cross-sectoral knowledge sharing, and a stronger research network capable of addressing compound event risks.

## 5 Recommendations for the compound event research community

Traditional natural hazard risk assessment and management approaches typically only consider one driver and/or hazard at a time, potentially leading to an underestimation of risk because the driver and hazards associated with extreme events often interact and are spatially and/or temporally dependent (Van den Hurk et al., 2023). Improving our understanding of compound events – and compounding risk – can provide a bridge between climate scientists, practitioners and policymakers who need to work closely together to understand, communicate, and manage the risks from these complex events. This review has documented the significant advancements in compound event research from 2012 to 2022, highlighting key opportunities for the next decade, as summarised in Fig. 4.




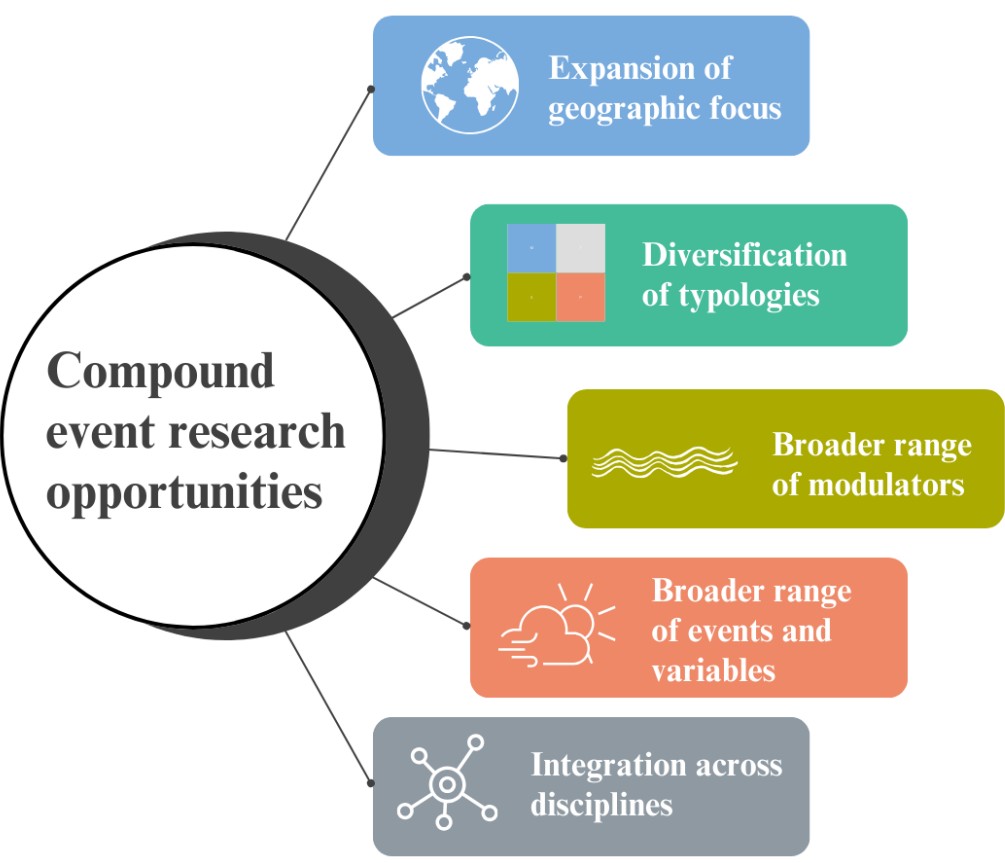

**Figure 4. Summary of potential future compound event research opportunities identified following the review process.**

Further research regarding compound events is pivotal to adapt to, and manage, increasingly severe extreme weather within
our changing climate. The opportunities outlined within this review include:

- **Expansion of geographic focus**: Current research is largely concentrated in Europe, Asia, and North America, creating a geographical bias. Expanding studies to other regions is essential for a more comprehensive global risk assessment. Utilising satellite data and remote sensing can help overcome data limitations in these areas, improving risk assessments and adaptation strategies.

- **Diversification of typologies**: The majority of existing studies have focused on multivariate (co-occurring) compound events. However, increasingly exploring pre-conditioned, temporally, and spatially compounding events is crucial for understanding complex interactions and enhancing risk assessments. This would lead to more effective risk management, early warning systems and adaptation strategies.





- **Exploration of modulators**: While some modulators, such as ENSO, have been studied extensively, there is limited understanding of how different modulators interact to influence compound events. Studying these interactions could enhance predictive models and early warning systems, particularly in sectors like agriculture, energy, and disaster risk management.

- **Wider range of events and variables**: Many compound events and hydrometeorological variables remain understudied, potentially leading to gaps in natural hazard risk assessments. Researching a broader range of events could improve predictive models, early warning systems, and climate adaptation strategies by accounting for a wider array of extreme weather scenarios.

- **Integration across disciplines**: Further integration of compound event research across disciplines and sectors is recommended. Standardising terminology and fostering collaboration will help capture the wide-ranging impacts of compound events, leading to more comprehensive risk assessments and cross-sectoral adaptation strategies.

Recent studies have already begun exploring these opportunities. For instance, Bastos et al. (2023) propose a systemic framework for analysing how extreme weather interacts with natural variability impact ecosystems, focusing on temporally compounding and pre-conditioned events. Ramos et al. (2023) examine the compound nature of the exceptional fires in Portugal in 2017, emphasising the role of multiple pre-conditioning factors. Markantonis et al. (2022) investigate past and future wet-cold compound events in Greece, while Sun et al. (2023) demonstrate how global warming increases the frequency and intensity of heatwaves combined with extreme precipitation and runoff. Sun et al. (2023) also highlights the potential impact of these compound events, showing that population exposure could more than triple by the end of the century under high-emission scenarios compared to lower-emission scenarios.

Compound event research is also advancing modelling approaches to capture the complexity of interactions between multiple climate drivers and hazards. For example, Bevacqua et al. (2023) emphasise the need for large ensemble simulations, specifically Single Model Initial-condition Large Ensembles (SMILEs), which provide extensive data spanning hundreds to thousands of years of simulated weather conditions. These large datasets are crucial for improving the reliability of climate risk assessments and projections. Furthermore, Nederhoff et al. (2024) introduces the Tropical Cyclone Forecasting Framework (TC-FF), a new method for probabilistically forecasting compound flooding caused by tropical cyclones. The method integrates key physical drivers like tide, surge, and rainfall, using Gaussian error distributions and autoregressive techniques to generate wind fields and produce probabilistic wind and flood hazard maps. Unlike traditional methods, TC-FF does not rely on detailed historical error distributions, making it adaptable for use in data-scarce regions like oceanic basins. Although this review did not focus on modelling approaches and methodologies employed within the compound event research reviewed between 2012-2022, papers such as Bevacqua et al. (2021) have highlighted useful methods for each compound event typology. Furthermore, **t**he continuously increasing amount of observed and modelled data on climate conditions and





impacts offers new opportunities for innovative data-driven approaches that quantify dependencies and identify relevant drivers. For example, recent studies have also exploited the abundance of simulated and observed data and used interpretable machine learning approaches to identify compounding drivers of forest mortality (Anand et al., 2024) and floods (Jiang et al., 2022, 2024).

Going forward, artificial intelligence (AI) will also play a crucial role in continually cataloguing, reviewing and documenting
the many advancements in compound event research. Ongoing documentation of published research can help the research community continue to address emerging opportunities and contribute valuable scientific outputs in the field, thereby enhancing adaptation to and mitigation of compound events moving forward.

## 6 Conclusions

The review highlights the substantial growth in research on compound weather events over the decade following the SREX
report (IPCC, 2012), which emphasised the need to better understand the interactions between multiple climate drivers and hazards that can intensify extreme weather impacts. The findings of this review indicate a significant expansion in the field, with the number of annual publications rising from fewer than 20 before 2018 to 116 between 2021 and 2022. Research has primarily focused on multivariate compound events, which account for 71.9% of studies, while other types, such as temporally compounding (11.9%) and spatially compounding events (3.6%), have been far less explored. ENSO is identified as the most
frequently analysed modulator, with many studies concentrating on compound hot and dry events and flood-related events, particularly regarding their impacts on flood/storm damage, agriculture, and ecosystem health.

Despite this progress, the review identifies several gaps and opportunities in compound event research. For instance, there is a need for more studies focusing on pre-conditioned, temporally, and spatially compounding events, which have received less attention compared to multivariate events. Additionally, research has been geographically skewed, with limited studies
addressing compound events in the Global South, where the impacts of such events can be particularly severe. Addressing the range of scientific opportunities outlined in this review will help researchers to continue to explore emerging compound event challenges and contribute valuable scientific outputs to the field. As extreme weather events continue to intensify, understanding and mitigating compound events is not only a scientific challenge but also a global imperative. This knowledge is crucial for safeguarding society and the environment against the escalating risks posed by compound weather and climate
events.



**Author contribution**

Conceptualisation: LB & CW; Methodology: LB & CW; Data curation: LB; Formal analysis: LB; Supervision: CW; Visualisation: LB; Writing – original draft preparation: LB; Writing – review & editing: all authors.

**Competing interests**

At least one of the (co-)authors is a member of the editorial board of Natural Hazards and Earth System Sciences.

**Acknowledgements**

LB was supported by the Engineering & Physical Sciences Research Council (EPSRC) [EP/T517938/1]. CJW acknowledges support from the NERC Global Partnerships Seedcorn Fund 'EMERGE' project though grant no. NE/W003775/1. CJW was also supported by the European Union's Horizon Europe 'Multi-hazard and risk informed system for enhanced local and
regional disaster risk management (MEDiate)' project under grant agreement no. 101074075. BvdH has received funding from the European Union's Horizon 2020 research and innovation programme (RECEIPT, grant agreement no. 820712). PJW has received funding from the European Union's Horizon 2020 research and innovation programme (MYRIAD-EU, grant agreement no. 101003276). Support from the Swiss National Science Foundation through project PP00P2_198896 to D.D. is gratefully acknowledged. This project has received funding from the European Research Council (ERC) under the European
Union's Horizon 2020 research and innovation programme (grant agreement No. 847456). JZ acknowledges the Helmholtz Initiative and Networking Fund (Young Investigator Group COMPOUNDX, Grant Agreement VH-NG-1537).

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
