# Peer review of "Review article: The growth in compound weather events research in the decade since SREX"

_Natural Hazards and Earth System Sciences, 2024_

## Referee Comment (RC2)

I would like to compliment the authors for a very well-written paper. This work makes a significant contribution to the literature by synthesizing a substantial body of research from the past decade, thereby highlighting key research gaps. Additionally, the paper offers concrete recommendations for future research, which will likely play an important role in shaping upcoming research agendas.

Like Reviewer 1, I found the distinction between multi-hazards and compound events somewhat unclear, particularly the statement that multi-hazards take a 'risk perspective.' Do the authors imply that compound event research does not account for risk components such as (dynamic) vulnerability and exposure, beyond the hazard itself? This seems inconsistent with the discussion in Section 3.3, where impacts, closely tied to risk and its components, are addressed. For example, the study by Tanir et al. (2021), cited in the manuscript, demonstrates that compound event research does consider risk. It may be worth further exploring the distinction between studies that incorporate risk and those that do not.

Small technical corrections:

L122: hazard **pairs** instead of **papers**.

L206-207: "17 types were mentioned  for their influence on the frequency and/or intensity of compound events across 12.2% of reviewed papers."

---

## Author Comment (AC1)

Dear Reviewer RC1,

Thank you for your constructive feedback and positive remarks. We are grateful for your detailed comments and agree that addressing these points will enhance the paper.

Here, we outline how we will revise the manuscript accordingly:

- We will clarify the distinction between compound events and multi-hazards, providing a more in-depth discussion on how compound events research concentrates on the analysis of hydrometeorological hazards, along with the associated exposure and vulnerability to these hazards, as opposed to broader geophysical or biological hazards.

- We are also happy to provide more detail regarding the purpose of the 55 'Other' papers, explaining why they are included. The 'Other' relevant research encompassed in this review includes the development of compound event frameworks, the conceptualisation of compound events within wider risk frameworks, and examples of compound weather and climate events within our conceptual analysis.

- We will address the four technical corrections highlighted by RC1 to clarify our findings (lines 121-122, 122-123, 213, and 227).

We value your perspectives and are confident that these adjustments will enhance the manuscript. Thank you once more for your time and expertise.

Best regards,
Lou (on behalf of the manuscript authorship team)
16th March, 2025

.

---

## Author Comment (AC2)

Dear Reviewer RC2,

Thank you for your positive comments and constructive feedback on our manuscript. We appreciate both your thorough review and your valuable insights into how we can improve the clarity and detail of our work.

Below, we outline how we intend to address the key concerns raised in the next phase of the review:

- We agree on the necessity of providing clarity regarding the distinction between multi-hazards and compound events, as well as the extent to which research on compound events considers risk components such as vulnerability and exposure, beyond just the hazard itself. While research on compound events can encompass considerations of vulnerability and exposure, we will elaborate on the overarching distinction that compound events research specifically focuses on combinations of hydrometeorological hazards. This is in contrast to other hazards, such as geophysical and biological hazards, which may still be included in multi-hazard research.

- We are also happy to clarify that our review focuses on analysis of pairs or combinations of hydrometeorological hazards. However, it still includes other relevant work related to compound events research, such as the development of compound event frameworks and examples of compound weather and climate events within conceptual analysis. These 'Other' papers are included in this review to highlight the theoretical development of compound events science. They are thus included in the supplementary information 1 reference list as relevant papers to explore.

- We will address the two technical corrections identified by RC2 by making the necessary amendments.

We appreciate your insights and believe that these revisions will significantly strengthen the manuscript. Thank you once again for your time and expertise.

Best regards,
Lou (on behalf of the manuscript authorship team)
16th March 2025

.

---

## Author Response (AR1)

**RC1**

Please note that our responses to the reviewer's comments are in red, while edits from the manuscript are in red and italics, with the line number referring to the first line of text copied.

The authors provide a comprehensive review of the growth of research in compound weather events. This paper is crucial because it compiles the existing work on compound events, a new area of research that has gotten significant traction in a short time. Moreover, a comprehensive review article on this component is rare. I believe the manuscript will attract researchers from different disciplines to provide valuable information about the recent developments in compound event research. Moreover, the authors identified the area where compound event research is limited and offer valuable and useful suggestions for future research opportunities.

Thank you for your supportive comments.

The authors briefly discuss compound events and multi-hazards in the introduction section. First, they mention that compound research shares similarities with the multi-hazard discipline. At the same time, later, they provide distinct differences between them: compound event research quantifies the interconnections between hazards/ drivers, while multi-hazards explore hazards from a risk perspective. However, there are studies in the literature where compound events research went beyond understanding or quantifying the interconnections and included risk estimates and societal impacts. It seems that these works combined compound events and multi-hazards. How can these works be distinguished in terms of compound events and multi-hazards discipline? Or should we name it as a separate typology of the research area? The authors might be interested in shedding more light on this discussion.

Thank you for this comment. We agree with your comment and have amended the text to clearly outline the similarities and differences between compound events and multi-hazards research as follows:

*L60: To help structure our thinking around the many possible types of compound events, Zscheischler et al. (2020) presented a typology for compound events comprising four compound event categories: multivariate, pre-conditioned, spatially compounding and temporally compounding (See Table 1 for definitions). Compound event research shares similarities with the multi-hazards discipline that explores "...the selection of multiple major hazards that a given country faces, and the specific contexts where hazardous events may occur either simultaneously, cascading, or cumulatively over time" (UNDRR, 2017). These similarities can often lead to the terms 'compound events' and 'multi-hazard events' being used interchangeably. However, while the multi-hazards discipline explores the interaction between a wide range of hazards – including hydrometeorological, biological, environmental, geological and technological processes and phenomena – compound events research is primarily motivated by the lack of consideration of compounding drivers in climate science risk assessments, with an ambition to support climate impact assessment or management (Van den Hurk et al., 2023). Compound events therefore refer to combinations of hazards and drivers related specifically to weather and / or climate (Zscheischler et al., 2018) and can be considered a subset of broader multi-hazard events. Additionally, given the complexity of multi-hazard research involving many potential combinations of hazards, there has only recently been much progress in the quantitative assessment of hazard interactions. For example, previous work has used simple single hazard layering approaches to provide a multi-hazard comparison, or qualitative / semi-qualitative approaches to examine the linkages between hazards (Ciurean et al., 2018), whereas compound events research has placed a stronger emphasis on considering the quantitative interdependencies between hazards (Tilloy et al., 2019). Furthermore, multi-hazards research primarily evolved from the disaster risk reduction (DRR)*

*field (Ward et al., 2022), whereas compound events research has its origins in climate impact research (Leonard et al., 2014). Consequently, although there are shared interests with the multi-hazards discipline, compound events research specialises in quantifying and understanding the interactions between multiple climate hazards, their drivers, and/or associated impacts, with the aim of better managing the effects of these hydrometeorological hazards on society (Tilloy et al., 2019; Simmonds et al., 2022; Van den Hurk et al., 2023).*

Line 121 – "55 did not quantitively analyse any compound events". So, what did the papers precisely do?

Please check lines 121-122.

Thank you for this comment. We appreciate this sentence ends abruptly. We have added extra detail to the previous paragraph outlining what the 'other' papers cover. We also added additional detail to the sentence queried by the reviewer to outline what the papers 'do' and why they are included in the review. Furthermore, we added a short paragraph in the results Section 3.1 to highlight the growth in the 'other' papers. Please see the three changes made below:

(1) Previous paragraph outlining what the 'other' papers cover:

*L131: Furthermore, theoretical research, review papers, conceptual frameworks, and other relevant papers (e.g., Raymond et al., 2020; Ebi et al., 2021; Zhang et al., 2021) that did not focus on place-specific analysis of compound events were classified as 'other'.*

(2) (Changes to the original comment raised by the reviewer:

*L135: Out of the 366 papers, several papers examined more than one compound event, such as Ridder et al. (2020) which studied 27 hazard pairs. Conversely, 55 'other' papers, including Zscheischler et al. (2020), which outlines a typology for compound events, and Gallina et al. (2016), which reviews methods for exploring compound events and emphasises the necessity for comprehensive multi-risk approaches, did not quantitatively analyse any compound events. However, these papers were included in the review to (1) illustrate the overall development of compound events research, including theoretical advancements over the 10-year period, and (2) to be catalogued in the supplementary material, serving as a valuable resource for policymakers, practitioners, and researchers in future access to compound events research. Consequently, in total, this review catalogued 388 compound events, many of which share similar hazard combinations (e.g., high temperature and low precipitation), from across the 366 papers reviewed.*

(3) Here is the paragraph added to the results section 3.1:

*L197: The annual number of published papers that did not include place-specific analysis (labelled as 'other' in Fig. 1) remained relatively low over the ten-year period, showing a gradual increase from one paper in 2012-2013 to seven papers in 2020-2021. This upward trend may be attributed to the publication of several review and perspective papers that address the challenges associated with researching compound events, such as those by AghaKouchak et al. (2020) and Raymond et al. (2020), as well as the development of conceptual frameworks such as Zscheischler et al. (2020).*

Lines 122-123, "this review catalogued 388 separate compound events". There are not 388 different types of compound events. Instead, the authors found 388 compound events, many of which are the same as it is they were categorized later, right?

Yes, thank you, this is correct. We have edited this sentence for clarity as follows:

*L141: Consequently, in total, this review catalogued 388 compound events, many of which share similar hazard combinations (e.g., high temperature and low precipitation), from across the 366 papers reviewed.*

Line 213—It is not clear in the text that the Jarvis et al. (2018) paper talks about the combined modulator (i.e., the co-occurrence of ENSO and IOD modulators is considered a compound event), which is different than the modulation of compound events' frequency and intensity by large-scale climatic modulators.

Thank you for this comment. We have added the words 'combined' and 'co-occurring' to make this clear:

*L236: ... and Jarvis et al. (2018) explore the combined influence of ENSO and the Indian Ocean Dipole (IOD) co-occurring on winegrape maturity in Australia.*

Line 227 – the authors might clarify between "mentioned" and "analyzed". Are there papers that fall into both categories?

Thank you for highlighting the need for clarity here. We have edited the methods to make this clearer:

*L155: These were then grouped into two categories: (1) modulators quantitatively analysed and (2) modulators mentioned but not analysed, with no papers falling into both categories for a given modulator. Modulators were further classified into (1) ocean warming patterns and (2) persistent atmospheric Rossby wave configurations to compare their frequency of mentions or analysis in the reviewed papers.*

**RC2**

Please note that our responses to the reviewer's comments are in red, while edits from the manuscript are in red and italics, with the line number referring to the first line of text copied.

I would like to compliment the authors for a very well-written paper. This work makes a significant contribution to the literature by synthesizing a substantial body of research from the past decade, thereby highlighting key research gaps. Additionally, the paper offers concrete recommendations for future research, which will likely play an important role in shaping upcoming research agendas.

Thank you for your positive comments.

Like Reviewer 1, I found the distinction between multi-hazards and compound events somewhat unclear, particularly the statement that multi-hazards take a 'risk perspective.' Do the authors imply that compound event research does not account for risk components such as (dynamic) vulnerability and exposure, beyond the hazard itself? This seems inconsistent with the discussion in Section 3.3, where impacts, closely tied to risk and its components, are addressed. For example, the study by Tanir et al. (2021), cited in the manuscript, demonstrates that compound event research does consider risk.

It may be worth further exploring the distinction between studies that incorporate risk and those that do not, and what risk means in the compound community.

Thank you for your questioning here. We agree that our efforts to outline the differences between compound events research and multi-hazards research was not as clear as it needed to be. Our edits outlined below now outline the differences between compound events and multi-hazards,noting that compound events research can include both impacts and a discussion of risk. Please see the below text:

*L60: To help structure our thinking around the many possible types of compound events, Zscheischler et al. (2020) presented a typology for compound events comprising four compound event categories: multivariate, pre-conditioned, spatially compounding and temporally compounding (See Table 1 for definitions). Compound event research shares similarities with the multi-hazards discipline that explores "...the selection of multiple major hazards that a given country faces, and the specific contexts where hazardous events may occur either simultaneously, cascading, or cumulatively over time" (UNDRR, 2017). These similarities can often lead to the terms 'compound events' and 'multi-hazard events' being used interchangeably. However, while the multi-hazards discipline explores the interaction between a wide range of hazards – including hydrometeorological, biological, environmental, geological and technological processes and phenomena – compound events research is primarily motivated by the lack of consideration of compounding drivers in climate science risk assessments, with an ambition to support climate impact assessment or management (Van den Hurk et al., 2023). Compound events therefore refer to combinations of hazards and drivers related specifically to weather and / or climate (Zscheischler et al., 2018) and can be considered a subset of broader multi-hazard events. Additionally, given the complexity of multi-hazard research involving many potential combinations of hazards, there has only recently been much progress in the quantitative assessment of hazard interactions. For example, previous work has used simple single hazard layering approaches to provide a multi-hazard comparison, or qualitative / semi-qualitative approaches to examine the linkages between hazards (Ciurean et al., 2018), whereas compound events research has placed a stronger emphasis on considering the quantitative interdependencies between hazards (Tilloy et al., 2019). Furthermore, multi-hazards research primarily evolved from the disaster risk reduction (DRR) field (Ward et al., 2022), whereas compound events research has its origins in climate impact research (Leonard et al., 2014). Consequently, although there are shared interests with the multi-hazards discipline, compound events research specialises in quantifying and understanding the interactions between multiple climate hazards, their drivers, and/or associated impacts, with the aim of better managing the effects of these hydrometeorological hazards on society (Tilloy et al., 2019; Simmonds et al., 2022; Van den Hurk et al., 2023).*

Small technical corrections:

L122: hazard **pairs** instead of **papers**.

Thank you. We have changed hazard papers to hazard pairs.

L206-207: "17 types were mentioned within reviewed papers for their influence on the frequency and/or intensity of compound events across 12.2% of reviewed papers." *within reviewed papers* can be removed from sentence.

Thank you. We removed 'within reviewed papers' from this sentence.